# Comparative Analysis of Platelet-Derived Extracellular Vesicles Using Flow Cytometry and Nanoparticle Tracking Analysis

**DOI:** 10.3390/ijms22083839

**Published:** 2021-04-07

**Authors:** Sobha Karuthedom George, Lucia Lauková, René Weiss, Vladislav Semak, Birgit Fendl, Victor U. Weiss, Stephanie Steinberger, Günter Allmaier, Carla Tripisciano, Viktoria Weber

**Affiliations:** 1Center for Biomedical Technology, Department for Biomedical Research, Danube University Krems, 3500 Krems, Austria; sobha.karuthedom@gmail.com (S.K.G.); lucia.krajcik-laukova@donau-uni.ac.at (L.L.); rene.weiss@donau-uni.ac.at (R.W.); vladislav.semak@donau-uni.ac.at (V.S.); birgit.fendl@donau-uni.ac.at (B.F.); carla.tripisciano@donau-uni.ac.at (C.T.); 2Institute of Chemical Technologies and Analytics, TU Wien, 1060 Vienna, Austria; victor.weiss@tuwien.ac.at (V.U.W.); stephanie.steinberger@tuwien.ac.at (S.S.); guenter.allmaier@tuwien.ac.at (G.A.)

**Keywords:** extracellular vesicles, flow cytometry, nanoparticle tracking analysis, phosphatidylserine, platelets

## Abstract

Growing interest in extracellular vesicles (EVs) has prompted the advancements of protocols for improved EV characterization. As a high-throughput, multi-parameter, and single particle technique, flow cytometry is widely used for EV characterization. The comparison of data on EV concentration, however, is hindered by the lack of standardization between different protocols and instruments. Here, we quantified EV counts of platelet-derived EVs, using two flow cytometers (Gallios and CytoFLEX LX) and nanoparticle tracking analysis (NTA). Phosphatidylserine-exposing EVs were identified by labelling with lactadherin (LA). Calibration with silica-based fluorescent beads showed detection limits of 300 nm and 150 nm for Gallios and CytoFLEX LX, respectively. Accordingly, CytoFLEX LX yielded 40-fold higher EV counts and 13-fold higher counts of LA^+^CD41^+^ EVs compared to Gallios. NTA in fluorescence mode (F-NTA) demonstrated that only 9.5% of all vesicles detected in scatter mode exposed phosphatidylserine, resulting in good agreement of LA^+^ EVs for CytoFLEX LX and F-NTA. Since certain functional characteristics, such as the exposure of pro-coagulant phosphatidylserine, are not equally displayed across the entire EV size range, our study highlights the necessity of indicating the size range of EVs detected with a given approach along with the EV concentration to support the comparability between different studies.

## 1. Introduction

EVs are subcellular fragments that originate from the endosomal compartment or are shed from the plasma membrane of virtually all cell types of the human body under both, physiological and pathological conditions [1,2,3]. They have been recognized as major players in intercellular communication and can exhibit variable functions, depending on their cellular origin, their membrane composition, and surface-associated proteins, as well as their cargo [4,5].

The generic term “extracellular vesicles” encompasses individual vesicle subpopulations, which display overlapping features, despite their heterogeneity [6,7]. Small EVs (exosomes; 40–100 nm) originate from endosomal multivesicular bodies, whereas the release of large EVs (microvesicles; 100–1000 nm) involves cytoskeletal contraction as well as a rearrangement of plasma membrane phospholipids, leading to the exposure of phosphatidylserine on the vesicle surface (Figure 1) [1,8].

Isolation and characterization of individual EV populations require a combination of different methods [6] and still remain challenging due to the submicrometer size of EVs and the heterogeneity of EV populations. Moreover, when it comes to the analysis of EVs in complex fluids, such as human whole blood or plasma, EV size and density overlap with other biological structures. Low density and high density lipoproteins, in particular, are often co-isolated with EVs [9].

Flow cytometry is well-established for the characterization of EVs directly in body fluids because of its high throughput and its ability to label and discriminate EVs of different cellular origin. Still, current flow cytometers are not capable of fully resolving individual EVs on the basis of light scatter [10], as cryo-electron microscopy demonstrates that over 75% of EVs are less than 500 nm in diameter. Moreover, swarm detection may lead to erroneous data interpretation in flow cytometry. Swarming occurs when the concentration of particles in a sample is so high that light scatter or fluorescent signals generated by individual events can no longer be separated from each other [11].

Despite these limitations, EV detection using flow cytometry is according to the current estimates used in 90% of all studies in the field of EV research [12,13], particularly as the resolution of conventional flow cytometers has steadily improved due to technical advancements. With this widespread use of flow cytometry, it has become apparent that the diversity in instrumentation and instrument settings has a large impact on the comparability of EV data between different flow cytometers.

NTA is suited for the rapid assessment of size and concentration of nanoparticles ranging from 40–1000 nm in diameter [14,15]. It combines laser light scattering with a charge-coupled device camera to trace the Brownian motion of particles in solution and calculates particle size according to the Stokes-Einstein Equation [16]. While NTA in scatter mode (S-NTA) is not able to discriminate EVs from other light-scattering entities, such as lipoproteins or protein aggregates, F-NTA can identify EVs after labelling with membrane dyes or with antibody-fluorochrome conjugates against specific EV surface proteins. Differential S-NTA/F-NTA can thus support the identification of EVs within heterogeneous samples [15,17].

Here, we employed flow cytometry and differential S-NTA/F-NTA for the characterization of different batches of platelet-derived EVs. Using two different flow cytometers (Gallios and CytoFLEX LX), we compared total EV counts as well as the amount of phosphatidylserine-exposing EVs and related these data to results obtained with differential S-NTA/F-NTA, revealing vast differences in EV counts for the different techniques and instruments.

## 2. Results

### 2.1. Characterization of EVs by Flow Cytometry

Eight batches of platelet-derived EVs from different donors were analysed on both, the Gallios and CytoFLEX LX flow cytometers. Flow cytometric analysis showed improved resolution of silica reference beads for CytoFLEX LX in comparison to the Gallios flow cytometer, with a detection limit of 150 nm and 300 nm, respectively (Figure 1D). Due to its ability to detect EVs down to the size range of exosomes, CytoFLEX LX yielded 40-fold higher EV counts (events/µL) and 13-fold higher counts of LA^+^CD41^+^ EVs/µL as compared to the Gallios flow cytometer (Figure 2A). LA^+^ events, i.e., EVs expressing phosphatidylserine, comprised 93% and 64% of all events in the EV gate for Gallios and CytoFLEX LX, respectively (Figure 2B). The vast majority of the LA^+^ events were also CD41^+^, confirming their platelet origin (Figure 2C).

### 2.2. Characterization of EVs by Nanoparticle Tracking Analysis in Scatter Mode

The same EV batches that were characterized by flow cytometry were analyzed by S-NTA to determine their size distribution and concentration (Figure 2D–F), yielding an average concentration of 4.0 ± 1.7 × 10^8^ particles per µL. Since NTA is not specific for EVs and detects any structure that scatters light, we refer to “particles” rather than “vesicles” in this context. The majority of particles measured in S-NTA were 150 nm in diameter, confirming that most EVs were in a size range below the detection limit of flow cytometry (Figure 2D).

### 2.3. Characterization of EVs by Nanoparticle Tracking Analysis in Fluorescence Mode

To differentiate particles containing a lipid membrane from non-lipid particles, we performed NTA in fluorescent mode after staining with the membrane dye CMO. In addition, samples were stained with LA-AF555 to label vesicles exposing phosphatidylserine, allowing for a direct comparison with flow cytometry. The majority of CMO^+^ and LA^+^ particles according to F-NTA were 250 and 370 nm in diameter, respectively (Figure 2D). Of all particles detected in S-NTA, 36% were CMO^+^, whereas only 9.5% were LA^+^ (Figure 2F). Comparing the results of lactadherin staining across the three instruments, both F-NTA and CytoFLEX LX detected an average of 3.3 × 10^7^ LA^+^ EVs/μL, while the counts of LA^+^ EVs/μL obtained with the Gallios flow cytometer were more than an order of magnitude lower (1.4 × 10^6^ LA^+^ EVs/μL, Figure 2A,E). Data on the characterization of identical samples by flow cytometry and differential S-NTA/F-NTA analysis are summarized in Table 1.

### 2.4. Characterization of EVs by Fourier-Transform Infrared Spectroscopy (FT-IR)

We employed FT-IR spectroscopy as additional label-free measurement mode to characterize EVs. Figure 3 shows a representative FT-IR spectrum for platelet-derived EVs. Individual spectra of eight EV batches are included in Appendix A. All FT-IR spectra were essentially identical. Minor variations in the ester peak (≈1740 cm^−1^) are likely due to donor-dependent variations in the concentration of blood lipids, e.g., levels of cholesterol and/or triglycerides. The FT-IR spectra for platelet-derived EVs were similar to previously published patterns for monocytic EVs [18] or red blood cell-derived EVs [19]. The strongest signals corresponded to peptide (–CO–NH–) backbone vibrations, i.e., amide A (3287 cm^−1^), amide I (1652 cm^−1^), and amide II (1546 cm^−1^). Peaks at 2923 and 2852 cm^−1^ corresponded to asymmetric and symmetric methylene (C–H) stretching. FT-IR spectroscopic protein-to-lipid ratios were calculated from the data according to Mihaly et al. [20] and yielded a mean protein-to-lipid ratio of 1.96 ± 0.54 (n = 8). Description of the most prominent FT-IR absorption bands of platelet-derived EV are summarized in Appendix A.

## 3. Discussion

In our current study, we compared the ability of two conventional flow cytometers and of nanoparticle tracking analysis to quantify platelet-derived EVs. NTA detects EVs down to a size of about 50 nm, whereas the flow cytometers used in our study, Gallios and CytoFLEX LX, have a detection limit of about 300 and 150 nm, respectively [3,8,21]. The lower detection limit of the CytoFLEX LX instrument is achieved by 405 nm (violet) instead of conventional 488 nm (blue) wavelength side scatter detection of EVs, since triggering on violet side scatter generates signals of significantly higher intensity. This increases the signal-to-noise ratio and consequently improves the resolution of smaller EVs over instrument noise [16,22]. For both devices, we used fluorescent silica particles as reference material, as their refractive index is closer to EVs as compared to polystyrene-based calibration beads (1.46 vs. 1.40 for platelet-derived EVs) [23].

In flow cytometry, we evaluated both, total EV counts (events/µL) and counts of EVs exposing phosphatidylserine, which were identified by staining with lactadherin-FITC. In agreement with its enhanced ability to detect smaller EVs, CytoFLEX LX yielded 40-fold higher total EV counts as compared to the Gallios instrument. Our study also revealed differences in the percentage of LA^+^ EVs between the two instruments. While more than 90% of all events detected in the EV gate with the Gallios instrument were LA^+^ and thus exposed phosphatidylserine on their surface, CyoFLEX LX identified only 64% of LA^+^ events in the EV gate. This is most likely explained by the fact that the detection of EVs using Gallios is limited to larger EV populations, which are preferentially derived from the plasma membrane and, in contrast to smaller EVs derived from the endosomal compartment, expose phosphatidylserine on their outer membrane leaflet. Our findings underline that reported EV concentrations should always be accompanied by information on the minimal detectable EV size for a given instrument. This is crucial to estimate the percentage of certain EV subpopulations, and is particularly essential wherever a functional trait (e.g., the exposure of pro-coagulant phosphatidylserine) is not uniformly distributed in a population, but is preferentially associated with vesicles of a particular size range (e.g., larger vesicles).

Comparing flow cytometry and NTA, the total events detected with NTA in scatter mode exceeded flow cytometric EV counts by more than one (CytoFLEX LX) or two (Gallios) orders of magnitude, respectively. This is in good agreement with previous data reported by van der Pol et al. who found 15 times lower EV counts with flow cytometry as compared to NTA [24], which was most probably due to (i) the different minimum detectable vesicle sizes for NTA vs. flow cytometry and (ii) the detection of non-EV light scattering structures, such as protein aggregates or lipoproteins, by NTA. To refine NTA analysis, we additionally used NTA in fluorescent mode after staining of EVs either with the unspecific intercalating membrane dye CMO or with fluorescently labelled lactadherin to detect EVs exposing phosphatidylserine. The latter approach allowed us to link NTA and flow cytometry data, and revealed a good agreement of LA^+^ EV counts for CytoFLEX LX (lactadherin-FITC) and NTA (lactadherin-AF555).

According to F-NTA analysis following labelling of EVs with CMO vs. lactadherin-AF555, 36% of all particles detected in scatter mode were stained with CMO, a fluorescent plasma membrane label composed of amphipathic molecules comprising a lipophilic moiety for membrane loading and a negatively charged hydrophilic dye for anchoring of the probe in the plasma membrane [25]. This indicates that the majority of particles detected with NTA in scatter mode are structures lacking a lipid membrane.

The smaller size of particles obtained by S-NTA as compared to F-NTA indicates the detection of smaller non-EV contaminants, such as protein aggregates or lipoproteins, in scatter mode. We are currently assessing whether further purification of EV samples by size exclusion chromatography results in the depletion of co-isolated protein complexes/lipoproteins, to reveal a more accurate size distribution of the “true” EV population in scatter mode. The increased size of LA^+^ particles in comparison to CMO^+^ particles supports the findings from flow cytometry that phosphatidylserine is predominantly present on larger EV populations. Still, EV sizes obtained with F-NTA might also be affected by the staining procedure itself, since antibody or dye bound to the EV surface can affect the Brownian motion of the particles [17].

While applying the same sample dilution for both, scatter mode and fluorescent mode would be optimal to avoid inaccuracies caused by pipetting, this approach is only applicable for EV samples derived from matrices with comparatively low complexity, such as EVs isolated from cell culture supernatants. In the case of complex sample matrices, such as platelet concentrate, however, measurements in scatter mode require a much higher dilution than in fluorescent mode due to the background caused by plasma protein aggregates or lipoproteins. To limit inaccuracies related to pipetting in our study, we performed all measurements at least in triplicates (i.e., three independent dilutions per sample).

As a supplement to our study, we evaluated the suitability of FT-IR spectroscopy to determine the protein-to-lipid ratio of platelet-derived EVs. Whereas FT-IR is not commonly applied for EV quantification, it has been used to characterize the composition of EVs of different cellular origin, e.g., prostate cancer cells [26], monocytic cells [18], Jurkat cells [20], or red blood cells [19]. Based on the ratio of the peak intensities of amide I (1650 cm^−1^) and C-H (2700–3000 cm^−1^) stretching vibrations, the spectroscopic protein-to-lipid ratio was proposed as sample quality parameter [20]. We obtained a protein-to-lipid ratio of 1.96 for platelet-derived EVs, which was considerably higher than the protein-to-lipid ratio reported for EVs derived from Jurkat cells (exosomes, 0.79; microvesicles, 0.60; apoptotic cells, 1.20) [20] or from red blood cells (1.3) [19]. However, at this stage, the significance of a direct comparison of these values is limited, as EVs were isolated from different matrices (cell culture medium vs. human plasma) and with different protocols in these previous studies, potentially yielding different amounts of co-enriched proteins or lipoproteins. Still, FT-IR spectroscopy—which requires a minimal amount of sample (8–10 µg protein) and no additional sample processing—could provide a useful approach for the screening of EV fractions. It could support the identification of impurities in EV samples by comparative analysis of EV fractions before and after the depletion of co-enriched proteins, for example by size exclusion chromatography.

It would be of interest to extend this study to other flow cytometers, particularly to instruments from other manufacturers, and to compare different NTA devices from ZetaView (PMX-110 vs. PMX-120 with higher sensitivity) as well as from other figureiers (e.g., NanoSight NS300, Malvern Instruments, Worcestershire, UK). Still, taking into account that pre-analytical parameters including storage of EV samples, particularly of platelet-derived EVs, induce additional EV release from residual platelets, we limited our investigations to the devices available in our laboratory, to avoid bias caused by sample storage and transportation.

The accurate characterization of EVs using well-defined protocols is crucial for subsequent functional studies, but the phenotypical characterization of the whole EV spectrum is difficult to attain. This is especially challenging for the characterization of EVs from highly complex matrices such as whole blood, which contain lipoproteins and protein aggregates with overlapping characteristics (size, density) [8,16,22]. While flow cytometry is one of the most versatile approaches for EV characterization, it faces limitations, such as the discrimination of small EVs from the instrument noise, precluding the analysis of EVs below 150 nm even with state-of-the-art instruments. Light scatter, on the other hand, becomes critically limited when analysing EVs with diameters below the wavelength of the detection light source [8,24,27]. Thus, it appears promising to combine techniques, using different and non-overlapping principles to further improve EV-characterization.

In conclusion, our study revealed profound differences in EV concentrations obtained with two different conventional flow cytometers and with NTA. Different vesicle concentrations were primarily caused by differences between the minimal detectable EV sizes in flow cytometry, and, additionally, by the detection of non-vesicular light scattering structures in NTA. Since certain functional characteristics, such as the exposure of pro-coagulant phosphatidylserine, are not equally displayed across the entire EV size range, our study highlights the necessity of indicating the size range of EVs detected with a given approach along with the EV concentration to support the comparability between different studies.

## 4. Materials and Methods

### 4.1. Enrichment of Platelet-Derived EVs

Medical grade platelet concentrates from healthy donors were provided by the Clinic for Blood Group Serology and Transfusion Medicine, Medical University of Vienna, Vienna, Austria, as approved by the local ethics committee (ECS2177/2015). Prior to sample acquisition, written informed consent was obtained from all donors. Samples were collected in a blood bank setting using a Trima Accel R automated blood collection system (Version 5.0, Terumo BCT, Lakewood, CO, USA). Platelet concentrates were stored in polyolefin bags in storage solution for platelets (SSP^+^; Macopharma, Tourcoing, France) at a ratio of 80% SSP^+^ and 20% plasma, and were used for EV isolation within 2 days.

EVs were enriched from platelet concentrates by differential centrifugation, as previously described [3]. Platelet concentrates were centrifuged at 2500× *g* for 15 min, at room temperature (RT) to deplete platelets and debris. EVs were pelleted at 20,000× *g* (30 min, 4 °C) using a Sorvall Evolution RC ultracentrifuge equipped with an SS-34 rotor (Thermo Fisher Scientific, Waltham, MA, USA). The pellet was washed with sterile phosphate buffered saline (PBS) without calcium and magnesium (Life Technologies, Paisley, UK), re-centrifuged at 20,000× *g* (30 min, 4 °C), and re-suspended in 200 μL PBS. The protein content was quantified using the DC Protein Assay (Bio-Rad, Hercules, CA, USA). Samples were normalized to a protein concentration of 4 mg/mL, aliquoted, and stored at −80 °C until further use. Eight batches from different donors were used for all measurements.

### 4.2. Flow Cytometric Characterization of Platelet-Derived EVs

EV suspensions were diluted in filtered PBS (0.1 µm Minisart syringe filter, Sartorius Stedim Biotech, Göttingen, Germany) to a protein concentration of 1 µg/mL. Aliquots of 100 µL of the diluted samples were stained for 15 min at RT in the dark with 83 ng fluorescein isothiocyanate-conjugated lactadherin (LA-FITC, Haematologic Technologies, Essex Junction, VT, USA) as marker of phosphatidylserine, as well as with 100 ng phycoerythrin cyanin 7 (PE-PC7)-conjugated anti-CD41 antibody (Beckman Coulter, Brea, CA, USA) as platelet marker. To avoid swarm detection, the optimal sample protein concentration was determined as specified in Appendix A. All antibody conjugates were centrifuged at 17,000× *g* for 10 min at RT prior to use to remove aggregates. Stained samples were diluted 5-fold in PBS and analyzed on a Gallios and a CytoFLEX LX flow cytometer (both from Beckman Coulter, Brea, CA, USA). All fluorochrome conjugates used for flow cytometry are specified in Appendix A. Fluorescent-green silica particles (1 μm, 0.5 μm, 0.3 μm, 0.1 μm; excitation/emission 485/510 nm; Kisker Biotech, Steinfurt, Germany) were used for calibration for both flow cytometers. The triggering signal was set to forward scatter for Gallios and to violet side scatter for CytoFLEX LX, and the EV gate was set as shown in Figure 1 [3,21]. Data were analyzed using the Kaluza Software (Beckman Coulter, Brea, CA, USA). For both flow cytometers, acquisition was performed for 3 min at a flow rate of 10 µL per minute, yielding the events per 30 µL. To calculate the number of events for each sample, the dilution factor during sample preparation and staining was taken into account.

### 4.3. Nanoparticle Tracking Analysis of Platelet-Derived EVs

EV counts and size distribution were assessed by NTA (ZetaView, PMX-110, Particle Metrix, Inning am Ammersee, Germany, equipped with a CCD camera, a 520 nm laser and a 550 nm long pass manual fluorescence filter). For measurements in scatter mode, isolated platelet-derived EVs were diluted 5000-fold (final concentration 0.8 μg protein/mL) in 0.1 μm filtered PBS. Measurements were performed in triplicates at RT at a camera sensitivity of 80%, counting an average of 1000 tracks with 15 frames per second. In addition to scatter mode measurements, fluorescence-based analysis was performed after staining with the membrane dye CMO (excitation/emission 554/567 nm, Invitrogen, Carlsbad, CA, USA) or with LA-AF555 (Haematologic Technologies, Essex Junction, VT, USA). All dyes and fluorochrome conjugates used for F-NTA are specified in Appendix A. Staining protocols were optimized by testing different EV-to-dye concentrations, as shown in Appendix A. For F-NTA, EV samples containing 10 µg protein were stained with either 20 ng CMO or with 660 ng LA-AF555 in a total volume of 22 μL PBS. Samples were incubated for 30 min at RT in the dark, and the stained EVs were diluted 100-fold or 55-fold in PBS for CMO staining and lactadherin staining, respectively, prior to analysis. Samples were analyzed at a camera sensitivity of 90%. Device calibration for scatter measurements was performed with NanoStandard (polystyrene standard 100 nm beads, Applied Micropheres, Leusden, The Netherlands) and with OR520 standard (100 nm, Particle Metrix, Inning am Ammersee, Germany) for fluorescence measurements. Data were analyzed using the ZetaView software version 8.04.02 (Particle Metrix, Inning am Ammersee, Germany). A correction factor was introduced to compare measurements recorded at different camera sensitivities (80% for scattering mode, 90% for fluorescence mode) [2].

### 4.4. Fourier-Transform Infrared Spectroscopy of Platelet-Derived EVs

All measurements were performed on a Spectrum Two FT-IR Spectrometer (PerkinElmer, Waltham, MA, USA) equipped with a LiTaO_3_ detector and a MIRacle™ single reflection Zinc Selenide ATR (ZnSe) accessory (PIKE Technologies, Madison, WI, USA). For spectral manipulations, the Spectrum 10 (PerkinElmer, Waltham, MA, USA) and OMNIC 8.1.0.10 (Thermo Fisher Scientific, Waltham, MA, USA) software versions were used. The detailed protocol for FT-IR measurements is described in the Text S1.

### 4.5. Statistical Analysis

Statistical analysis was performed using GraphPad Prism, version 8.2 (La Jolla, CA, USA). Data are presented as mean ± standard deviation (SD). One-way repeated measures ANOVA or two-way repeated measures ANOVA followed by Sidak’s multiple comparisons test were used to compare three groups. Statistical significances between two groups were determined by the paired *t*-test.

## Figures and Tables

**Figure 1 ijms-22-03839-f001:**
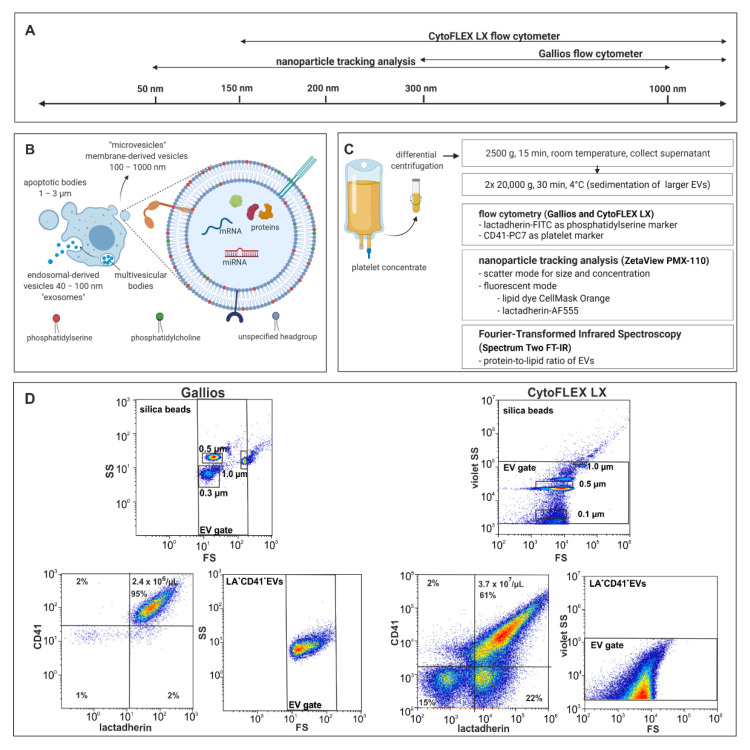
Characterization of EVs by flow cytometry and NTA. (**A**,**B**) Size-dependent resolution limits of the devices and approximate size range of EV subpopulations (exosomes, microvesicles, and apoptotic bodies). (**C**) Enrichment and characterization of platelet-derived EVs. Platelet-derived EVs were enriched from medical grade platelet concentrates by differential centrifugation as described in the Methods section and characterized by flow cytometry using lactadherin as a marker of phosphatidylserine expressing EVs and CD41 as platelet marker. Further analysis was performed by NTA in scatter mode and in fluorescence mode after staining of EVs with CellMask™ Orange (CMO) and lactadherin-Alexa Fluor™ 555 (LA-AF555). The protein-to-lipid ratio was assessed by Fourier-transformed infrared spectroscopy. (**D**) Representative images of EV gating and scatter plots for the CytoFLEX LX vs. Gallios flow cytometers. Phosphatidylserine exposing platelet-derived EVs were identified as lactadherin^+^ and CD41^+^ events in the EV gate. Figure 1A–C was created with BioRender.com (accessed on 5 April 2021).

**Figure 2 ijms-22-03839-f002:**
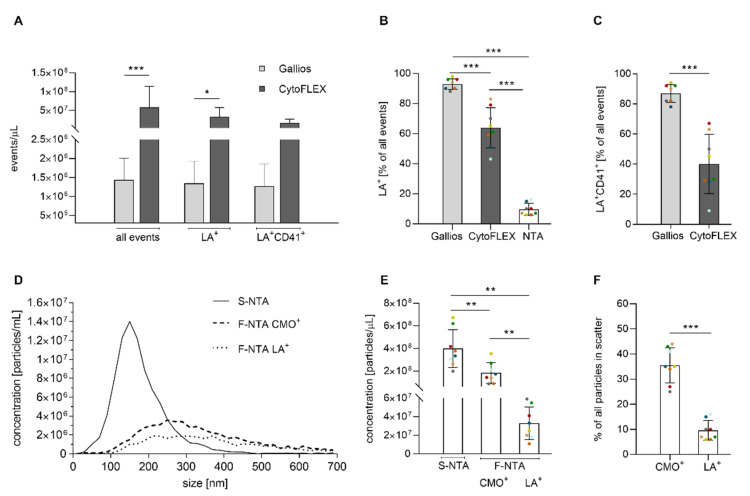
Comparative analysis of platelet-derived EVs using flow cytometry and NTA. (**A**) number of events detected by flow cytometry with Gallios vs. CytoFLEX LX; (**B**) percentage of LA^+^ events detected by flow cytometry using Gallios or CytoFLEX LX vs. percentage of LA^+^ particles detected by NTA in fluorescence mode; the percentages refer to the entirety of events detected with flow cytometry and NTA, respectively; (**C**) percentage of LA^+^CD41^+^ events for Gallios vs. CytoFLEX LX; (**D**) size distribution of platelet-derived EVs determined by NTA. Size distribution of EVs determined by S-NTA and F-NTA after staining with the unspecific membrane dye CMO (F-NTA CMO^+^) or with LA-AF555 (F-NTA LA^+^) as marker for phosphatidylserine exposing vesicles; (**E**) particle concentration detected by S-NTA and F-NTA (CMO^+^ and LA^+^); (**F**) percentage of CMO^+^ and LA^+^ particles detected by F-NTA. Data are presented as mean ± SD (n = 8, same batches used for all measurements; * *p* < 0.05; ** *p* < 0.01; *** *p* < 0.001).

**Figure 3 ijms-22-03839-f003:**
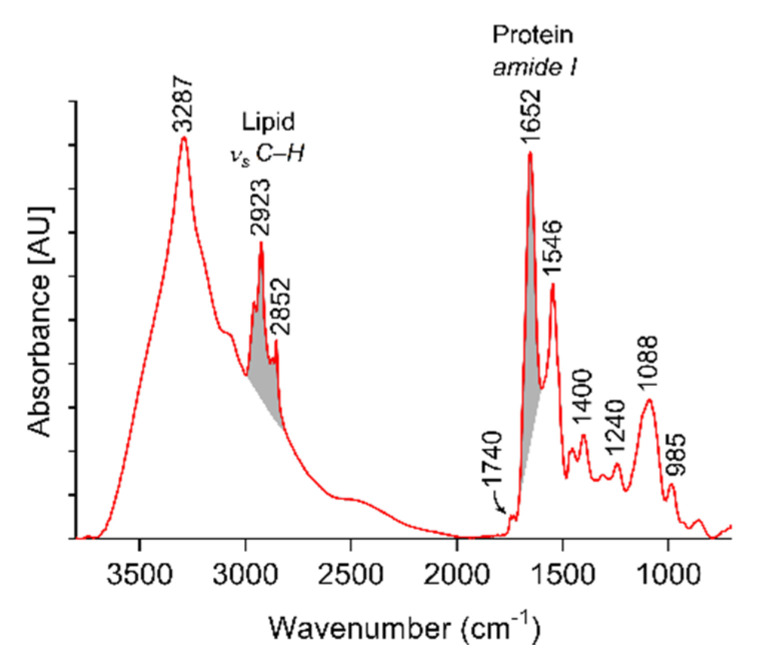
Characterization of platelet-derived EVs by FT-IR. Representative ATR/FT-IR spectra of EVs enriched from platelet concentrate. C–H (2700–3000 cm^−1^) and amide I (1600–1700 cm^−1^) stretching regions are highlighted in grey. AU, arbitrary units.

**Table 1 ijms-22-03839-t001:** EV counts obtained for platelet-derived EVs by flow cytometry (Gallios vs. CytoFLEX LX) and by NTA (scatter mode vs. fluorescence mode after staining with CMO or LA-AF555).

Flow Cytometry					
Device	Events/µL	LA^+^ EVs/µL	LA^+^ EVs [% of All Events in the EV Gate]	LA^+^CD41^+^ EVs/µL	LA^+^CD41^+^ EVs [% of All Events in the EV Gate]
Gallios	1.5 ± 0.5 × 10^6^	1.4 ± 0.6 × 10^6^	93 ± 4	1.3 ± 0.5 × 10^6^	87 ± 6
CytoFLEX LX	6.0 ± 5.5 × 10^7^	3.3 ± 2.5 × 10^7^	64 ± 13	1.7 ± 0.9 × 10^7^	40 ± 20
**Nanoparticle Tracking Analysis**					
ZetaView PMX-110	**Particles/µL** **[Scatter Mode]**	**LA^+^ Particles/µL** **[Fluorescent Mode]**	**LA^+^** **[% of All Particles in Scatter]**	**CMO^+^ Particles/µL** **[Fluorescent Mode]**	**CMO^+^** **[% of All Particles in Scatter]**
	4.0 ± 1.7 × 10^8^	3.3 ± 1.7 × 10^7^	9.5 ± 4	1.8 ± 0.9 × 10^8^	36 ± 7

n = 8; same EV batches used for all measurements.

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
