# Peer review of "Comparative Analysis of Platelet-Derived Extracellular Vesicles Using Flow Cytometry and Nanoparticle Tracking Analysis"

_ijms, 2021, doi:10.3390/ijms22083839_

Round 1

Reviewer 1 Report

the introduction provides a clear and to-the point overview of the technological status, and the shortcomings of both flow cytometry and nanoparticle tracking analysis. This is very concise and appropriate to the technical nature of the manuscript, without leaving biological consequences completely out of the equation. 

The choice of instruments fro flow cytometry (e Gallios and CytoFLEX LX flow cytometers) is probably inspired by availability in the laboratory - and thus justified. Nevertheless, it would be interesting to see a discussion how these 2 instriments would compare to other, probably more widely used flow cytometers as others may be interested in recapitulating and reporducing these data with THEIR instruments. Even better, it would have been brilliant to run some experiments in another lab, to further expand that "spectrum" of 2 instruments to a somewhat larger scope (in fact, both are from Beckman-Coulter; how would other makers of instruments perform in comparison?). 

This would be particularly interesting, as there is growing interest in the routine detection of exosomes/EVs by FACs, or other techniques, and a small but growing number of publications to achieve this. (For example, a recent paper in "Scientific Reports" on the issue to highlight the urgency: Scientific Reports | (2019) 9:2042 | https://doi.org/10.1038/s41598-019-38516-8 which is not even cited here). A simple google search further reveals a number of articles in sources such as "biocompare" or others that also show high interest in this issue from companies such as Becton-Dickinson, Miltenyi, or Thermo Fischer. Also the comparison to NTA might benefit from this, and vice versa, as this appears a valid approach. 

Otherwise, the different labeling technologies, specifically those used for S- and F-NTA, all appear highly relevant and (in my opinion) represent the moest critical component of the manuscript. But it also remains incomplete. Also here, additional comparisons may add to the story: for example in Fig. 3, how would different sources of EVs, and different preparation techniques, impact on the spectra? There are no comparisons shown, only 1 spectrum which we do not know if it is representative or not. 

It may also be interesing, for example, to discuss the impact of the methods introduced and compared here on the development of future exosome/EV assays, and standards required towards that goal. urthermore, an interesting, related aspect may be the iimpact of exosome isolation and analysis kits (commercially available and widely used) on the performance of cytometers and NTA. How would the methods compared here contribute to better characterizing and and analysing exosomes, or external proteins on exosomes, and how could it help to assess the purity of exosomes from different sources (cells, tissues) as this may be interesting to many readers... 

Reviewer 2 Report

The authors present a well designed and nicely presented study comparing the quantification of EVs (or rather of only the larger subpopulation of EVs - the MVs) by two conventional flow cytometers with different resolution and fluorescence-NTA. The novelty is high, since it is, as far as I know, one of the first attempts to directly compare EV analysis by FC and NTA. However, there are still some major points that need to be further improved in the manuscript.

Methodological concerns:

  1. The authors calculate the % of fluorescent CMO+ and LA+ particles in fl-NTA relative to all particles measured in scatter mode Since the samples in both modes are measured separately and their protein content and especially dilutions differ enormously, in my opinion it is not justified and feasible to directly compare both samples and calculate the % (especially the high dilution in scatter can lead to pipetting errors and inaccuracy). My lab is also using the ZetaView for fl-NTA and we managed to match the staining conditions in such way that  a measurement of the very same sample injected into the instrument first in fluorescent and then in scatter mode is possible. If this was not possible in the hands of the authors', they should at least explain it.
  2. 2 Fig1D: It si not clear if beads or EVs are shown in the FC/SSC plots If EVs are shown, then it should be explained that the size gates are set based on beads If beads areshown, then additional plots with EVs should be included.. For a better comparison of the 2 cytometers, the scatter plots of each should have the same orientation (FS vs. SS). 
  3. How the exact numbers of particles in FC are calculated when no counting beads are used? Are the numbers exact enough, taking into account the differences between the instruments (sample void volume, flow speed, etc.)?
  4. Fig2D: It would be interesting to see the size distribution also of  the CMO+ and LA+ particles to support the statements in lines 183-192 and 237-241, which lack enough evidence. The same in case of FC - backgating from the LA+CD41+ population to FS/SSC would show the size of these particles.
  5. In lines 194-196 the cytometers are inverted
  6. lines 221-224: explain what the higher protein/lipid ratio means in more detail (protein contamination?). What in comparison are the ratios of lipoproteins or the cell membrane?

Round 2

Reviewer 2 Report

The authors have adequately responded to my suggestions and the manuscript has improved a lot. However, in my opinion the manuscript would further benefit from placing suppl Fig. S3 within the main manuscript, e.g. in Fig. 2 next to the scatter NTA graph, since it contains important results, which should be also discussed more deeply (so far there was not even a reference to the suppl Fig in the text, so the newly added results escapes one's notice). First, the results indeed support the authors' statement that their isolation method preferentially separates larger LAC+ MVs, however the big size differences between the scatter and fluorescence mode are striking and should be discussed. I would expect that at least the CMO+ curve should cover also smaller particles and overlap more with the scatter curve. It would imply, that indeed the scatter measurement detects predominantly protein aggregates or lipoproteins - the authors mention it in the discussion, but I would stress it even more and discuss how other EV isolation methods (e. g. SEC) could improve the results obtained by NTA (or FC).
